# Using Drones to Determine Chimpanzee Absences at the Edge of Their Distribution in Western Tanzania

**Serge A. Wich** [1,*] , **Noémie Bonnin** [1] , **Anja Hutschenreiter** [2,3] , **Alex K. Piel** [4] , **Adrienne Chitayat** [5] , **Fiona A. Stewart** [1,4] , **Lilian Pintea** [6] **and Jeffrey T. Kerby** [7,8]

1   School of Biological and Environmental Sciences, Liverpool John Moores University, Liverpool L3 3AF, UK
2   Instituto de Investigaciones en Ecosistemas y Sustentabilidad, Universidad Nacional Autónoma de México, Morelia 04510, Mexico
3   ConMonoMaya A.C., Chemax 97770, Mexico
4   Department of Anthropology, University College London, London WC1H OBW, UK
5   Institute of Biodiversity and Ecosystem Dynamics, University of Amsterdam, 1012 WX Amsterdam, The Netherlands
6   Department of Conservation Science, The Jane Goodall Institute, Washington, DC 20036, USA
7   Aarhus Institute of Advanced Studies, Aarhus University, 8000 Aarhus, Denmark
8   Section for Ecoinformatics and Biodiversity, Department of Biology, Aarhus University, 8000 Aarhus, Denmark
*   Correspondence: s.a.wich@ljmu.ac.uk

**Abstract:** Effective species conservation management relies on detailed species distribution data. For many species, such as chimpanzees (*Pan troglodytes*), distribution data are collected during ground surveys. For chimpanzees, such ground surveys usually focus on detection of the nests they build instead of detection of the chimpanzees themselves due to their low density. However, due to the large areas they still occur in, such surveys are very costly to conduct and repeat frequently to monitor populations over time. Species distribution models are more accurate if they include presence as well as absence data. Earlier studies used drones to determine chimpanzee presence using nests. In this study, therefore, we explored the use of drones to determine the absence of chimpanzee nests in areas we flew over on the edge of the chimpanzee distribution in western Tanzania. We conducted 13 flights with a fixed-wing drone and collected 3560 images for which manual inspection took 180 h. Flights were divided into a total of 746 25 m² plots for which we determined the absence probability of nests. In three flights, we detected nests, in eight, absence was assumed based on a 95% probability criterion, and in two flights, nest absence could not be assumed. Our study indicates that drones can be used to cover relatively large areas to determine the absence of chimpanzees. To fully benefit from the usage of drones to determine the presence and absence of chimpanzees, it is crucial that methods are developed to automate nest detection in images.

**Keywords:** drone; UAV; chimpanzee; species conservation management

## 1. Introduction

A crucial aspect of conservation is using accurate and precise data to determine species distribution and density. Traditionally, these data have been collected using a number of ground-based methods such as line and point transects, reconnaissance surveys, camera traps, and passive acoustic recording units, all developed with an array of analytical methods [1–11]. In addition to these ground-based methods, aerial methods to count animals or their signs for distribution or density estimates using manned aircraft have been widely used (e.g., [12]. Aerial data collection to determine animal presence with drones equipped with various camera types (e.g., visual spectrum, thermal infrared) has become increasingly common and has been used to study a wide variety of terrestrial and marine species [13–15]. Typically, the aim is to directly detect animals to determine their presence and, in some cases, derive animal densities from those data (review in [13]).

For species occurring at low density and/or in environments where direct detection is challenging, researchers have relied on indirect signs of animal presence, such as nests in the case of great apes, as these are more abundant than the animals themselves and thus more easily detectable [16,17]. Ground-based surveys of nests as presence points allow inferences into distribution and density, as well as habitat use [17–22]. In addition to ground surveys to detect nests, aerial surveys with crewed helicopters have been used to count Bornean orangutan (*Pongo pygmaeus*) nests [12]. Due to the high costs, limited availability, and risks of crewed helicopters, alternative less costly and safer methods are needed. Multirotor and fixed-wing drones have recently been used to count orangutan (*Pongo* spp.) and chimpanzee (*Pan troglodytes*) nests [23–26].

So far, efforts to use drones have focused on determining whether ground survey data on chimpanzee nest detections correspond with detections from the air and thus on presence data. These studies have indicated that drone nest counts reliably predict the number of nests counted from the ground by researchers [23,24]. Such data are valuable for species distribution models, and presence-only methods that generate pseudo-absences from the background have been widely applied for numerous species distribution models (e.g., [27–30]. However, several species distribution models benefit from true absence data rather than background or pseudo absences [31,32] as these can improve model performance [30,33]. In particular, the use of true absence records is the only way to evaluate whether population estimates and trends are biased by trends in detection probability and, therefore, lead to enhanced inferential and predictive accuracy of species distribution models [31,34]. Such absence data have been collected for a number of species and survey methods [35–37]. We aimed to explore the applicability of drones to estimate presences and absences. We focused on chimpanzees because this species has a wide range spanning multiple countries, occurs in a mosaic of habitats, and occurs at low densities [38]. We conducted our survey at the easternmost edge of their geographical distribution. Chimpanzees in Tanzania occur in the western part of the country, although much of what is known about Tanzania's chimpanzees comes from two long-studied populations found in two National Parks (Gombe Stream and Mahale Mountains). Despite this, approximately 75% of Tanzania's 2-3000 chimpanzees occur outside of these National Parks [39–42]. Given the vast areas and the high costs of ground surveys to obtain data on chimpanzee presence and absence, it is important to explore alternative opportunities with drones that can cover large areas in short time frames and provide useful data. In this study, we aimed to determine the presence and absence of chimpanzee nests across their assumed eastern distributional limit in Tanzania through the use of drones with a standard optical camera.

## 2. Methods

We collected data for this study in western Tanzania in May 2016. Flights (n = 13) were conducted at three locations (Figure 1) across an area of approximately 525 km². The study area consists of a mosaic of evergreen forests, miombo woodlands, and grasslands that characterizes much of the chimpanzee distribution in Tanzania [29,42].

We used a fixed-wing Skywalker airframe equipped with a PixHawk flight controller in combination with a Sony DSC-RX100 II visual spectrum camera. The Pixhawk autopilot system included a computer processor, GPS, data logger, pressure and temperature sensor, airspeed sensor, triple-axis gyro, and accelerometer. We triggered the camera automatically based on a predefined flight plan to produce at least 60% front- and side-overlap images. We programmed the missions using the open-source software Mission Planner (http://planner.ardupilot.com/, accessed on 1 January 2015) on a laptop running the Windows operating system and conducted two types of flight missions. These were either simple linear transect flights or gridded missions in which more than 60% overlap between photos of parallel flight lines was required to cover a certain area completely (Table 1). Linear missions were better suited to follow riparian forests where chimpanzees tend to occur more than in the miombo woodlands. Grid missions were better suited for large forest blocks in which chimpanzees could occur. We flew the drone at various altitudes above

ground level as a result of undulating terrain and had varying total ground coverage as a function of varying flight heights, mission type, and duration (Table 1). A total of 3560 images were acquired during these flights (Table 1). JTK and SAW initially examined all images for the presence of chimpanzee nests, which took approximately 60 h for SAW and 30 h for JTK. SW made the final selection of nest images based on a second examination of the images, which took another 60 h.

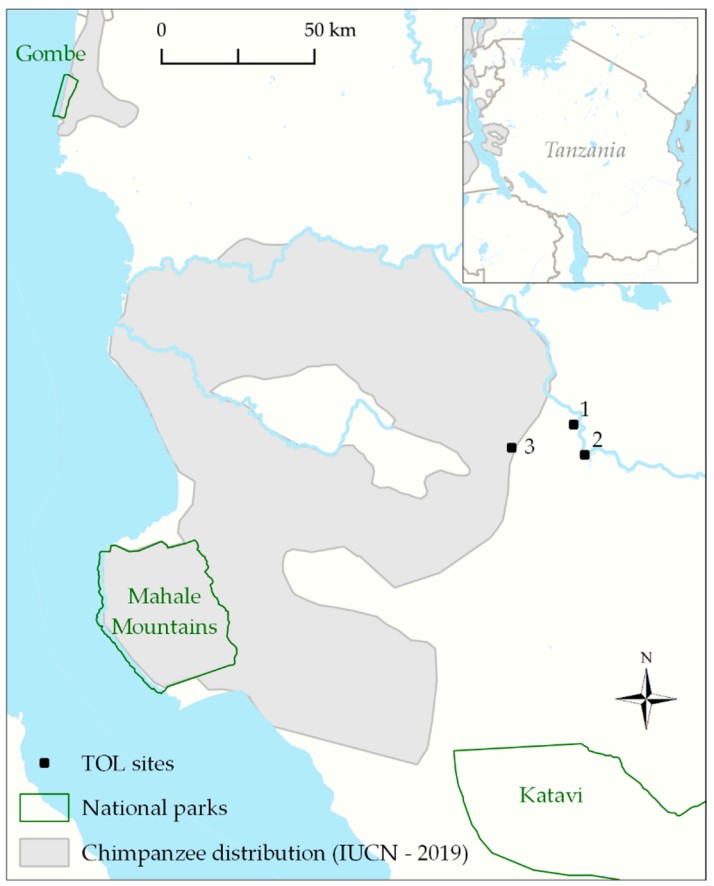

**Figure 1.** Map of the western Tanzania region in which the flights were conducted. TOL = take-off and landing.

**Table 1.** Results from N-mixture models using data from Bonnin et al. (2018).

| Rank | Model | AICc | ΔAICc | Detection $p$ (SE) | | Abundance $\lambda$ (SE) | |
| --- | --- | --- | --- | --- | --- | --- | --- |
| 1 | $\lambda (\cdot)\, p$ (GSD) | 162.98 | 0.000 | 0.196 | (0.045) | 39.6 (15.1) | |
| 2 | $\lambda (\cdot)\, p (\cdot)$ | 198.25 | 35.27 | 0.031 | (0.009) | 46.7 (14.2) | |

Notes. Nest abundance was modelled using a negative-binomial distribution of count data and a binomial distribution of nest detection probability depending on GSD reported in Bonnin et al. (2018). ΔAIC for the null model is >7 indicating that the selected observation-level covariate explains substantial variance in nest detection probability. The global model demonstrated good fit ($p = 0.325$) as tested with a Chi-squared statistic using the parboot function in the R package unmarked [43]. Average detection probability and standard errors were back-transformed from the logit scale to the original scale using the predict function in the R package unmarked. The negative-binomial error distribution was selected as it showed better fit than a Poisson error distribution and provided realistic abundance estimates [44].

We calculated the flight altitude above ground level for each image by subtracting the ground altitude (using a 30 m resolution Shuttle Radar Topographic Mission (SRTM); http://earthexplorer.usgs.gov, accessed on 1 April 2017 from the absolute flight altitude at which each image was taken which was obtained from the flight log and used to geotag each image using the Mission Planner software. For each plot, we checked whether nests

were detected in the images. We then calculated the number of plots surveyed on each mission (i.e., the number of spatial replicas per mission) by determining the total ground area covered by the images on each flight. We calculated this using Wimuas, which is a software that allows for the calculation of the covered area based on the images obtained during a flight [45]. Because flights had overlap in area coverage due to the take-off areas for several flights being similar or crossing flight paths, we only used the lowest flight for each overlapping area for analyses. Total ground area covered by the images was divided by 0.0025 km$^2$. We then calculated the probability of a false negative detection for each mission during which no nests were detected [37,46].

## 3. Results

To estimate the absence probability per mission, we used the formula developed by [37,46] that allows for the probability of false negative detections to be calculated based on $\alpha = (1 - p)^N$, where $p$ = detection probability during a survey and N = the number of survey replicas (Table 2). We used spatial replicates as the number of plots surveyed during a mission instead of temporal replicas (cf. space-for-time substitution [47]) for two reasons. First, nests are stationary and temporal replicas would not provide changes in availability unless over monthly or annual time-scales, which is essential to estimate the detection error [48]. Second, chimpanzee home ranges are larger than the area covered during any single drone flight (Nakamura et al. 2013), meaning that nest detection within one plot indicates the presence of a chimpanzee community in the whole area surveyed per mission. Plots measured 0.0025 km$^2$ based on similar-sized plots used in Bonnin et al. (2018). During this previous study, it was established that the higher a drone flight was above ground level (and hence the larger the ground sampling distance (the distance between the centres of two adjacent pixels in an image), GSD), the lower the resulting detection probability for a nest per 0.0025 km$^2$ plot [23]. This is due to nests becoming less clear in the images (Figure 2). We therefore used data from Bonnin et al. (2018) from repeated flights to run N-mixture models estimating chimpanzee nest detection probability depending on the GSD as an observation-level covariate (Table 1). We used the same relationship found between GSD and chimpanzee nest detection probability in the Bonnin et al. study to predict the nest detection probability per plot in the present study (Figure 3).

**Table 2.** Drone missions with flight characteristics and resulting probability to which real absence of chimpanzee nests can be assumed.

| Flight Number | Number of Photos | TOL | Alt AGL (m) | GSD (cm) | Mission Type | Area Covered (km$^2$) | # of Plots | Plots with Number of Nests | Detection Probability | Probability of a False Negative | Can Real Absence Be Assumed? (95% Probability) |
|---|---|---|---|---|---|---|---|---|---|---|---|
| 1 | 194 | 1 | 115 | 2.2 | grid | 0.87 | 19 | 0 | 0.098 | 0.141 | no |
| 2 | 333 | 1 | 177 | 3.4 | grid | 2.06 | 61 | 0 | 0.07 | 0.012 | yes |
| 3 | 183 | 1 | 190 | 3.7 | transect | 1.74 | 60 | 0 | 0.065 | 0.017 | yes |
| 4 | 203 | 1 | 305 | 5.9 | grid | 4.35 | 140 | 0 | 0.032 | 0.011 | yes |
| 5 | 50 | 1 | 100 | 1.9 | transect | 0.4 | 15 | 0 | 0.109 | 0.177 | no |
| 6 | 344 | 1 | 153 | 2.9 | transect | 2.62 | 102 | 0 | 0.080 | <0.001 | yes |
| 7 | 230 | 1 | 138 | 2.7 | transect | 1.71 | 66 | 0 | 0.088 | 0.002 | yes |
| 8 | 341 | 2 | 192 | 3.7 | transect | 4.81 | 192 | 0 | 0.064 | <0.001 | yes |
| 9 | 384 | 3 | 108 | 2.1 | transect | 1.64 | 62 | 0 | 0.104 | 0.001 | yes |
| 10 | 395 | 3 | 83 | 1.6 | transect | 1.23 | 29 | 0 | 0.12 | 0.024 | yes |
| 11 | 484 | 3 | 76 | 1.5 | transect | 1.17 | 46 | 1, 1 | 0.125 | 0 | (true positive) |
| 12 | 330 | 3 | 102 | 2.0 | transect | 1.29 | 50 | 1 | 0.108 | 0 | (true positive) |
| 13 | 89 | 3 | 142 | 2.7 | transect | 0.95 | 29 | 4, 4, 3, 1, 1, 2 | 0.086 | 0 | (true positive) |

Note: In the column 'Plots with number of nests', the comma-separated values indicate the number of nests found in unique plots. AGL = above ground level, GSD = ground sampling distance.

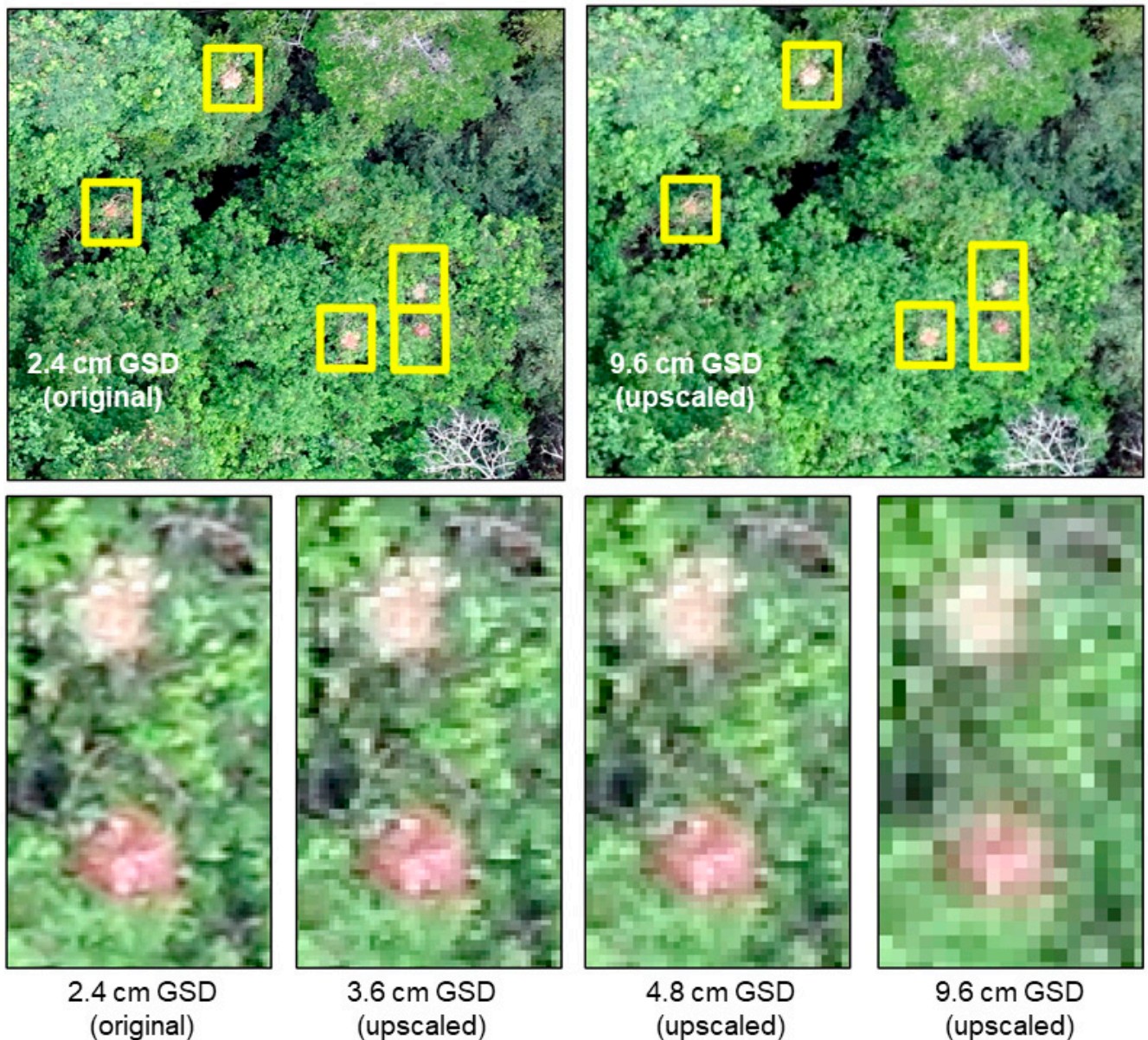

**Figure 2.** Simulated impacts on ground sampling distance (GSD) with increasing flight altitude or using a camera with a shorter focal length. Original imagery of nest IDs is presented next to this imagery upscaled to increasingly coarse GSD. Nest detection can be visually challenging even for trained observers. While potential nests may be visible in imagery with near decimetre GSD, validation of these as 'true nests' is extremely challenging, even when nests are located on a contrasting background like that pictured above. Many nests are likely missed due to being located in difficult-to-interpret contexts, and GSD likely impacts both initial detection and validation probabilities. To make this figure, we imported the original scaled image into ArcMap and exported it with pixel size multipliers of 1.5, 2, 3, and 4.

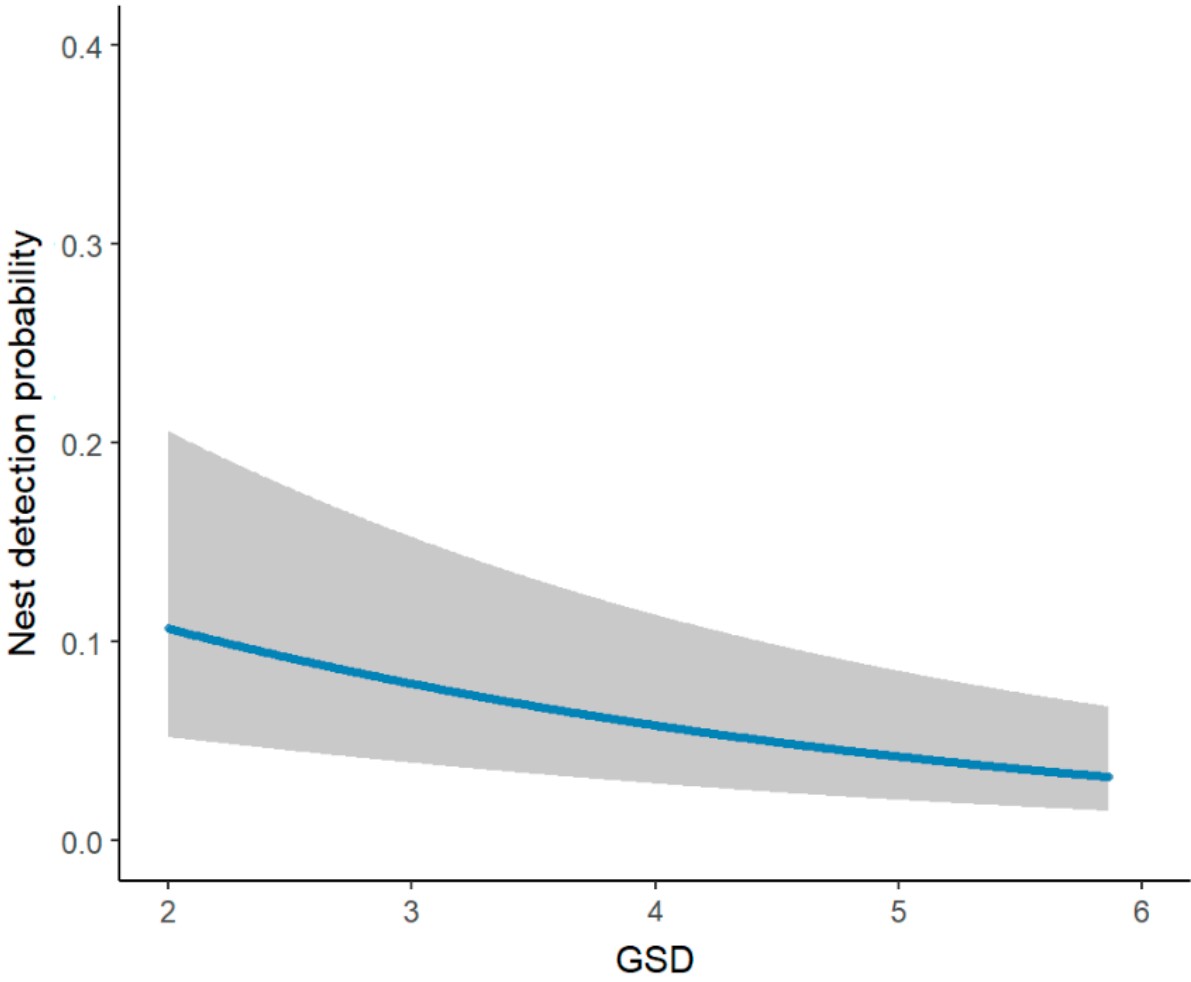

**Figure 3.** Predicted nest detection probabilities per 0.0025 km$^2$ plot depending on GSD (cm) based on N-mixture models using data from (2018). Grey lines represent 95% C.I. Detection probability and standard errors were back-transformed from the logit scale to the original scale and visualized using the predict function in the R package unmarked.

Of the 13 flights, nests were only observed in the TOL 3 area (Figure 4). In this area, 18 nests were observed on three different missions (Flights 11–13: Table 1). In flight 12, only one nest was observed in one plot, whereas in flight 11, there were two plots, each with one nest. Flight 13 contained the largest number of nests (N = 15) in the largest number of plots (N = 6: Table 2). The probability of a false negative in the plots where no nests were observed varied from <0.001 to 0.177 (Table 2). Using a 95% probability criterion (Figure 5), this indicated that in eight out of the ten flights in which no nests were detected, these absences fit the criterion.

It is important to assess whether drone surveys are more efficient than ground surveys, as measured by 'person days'. The total area covered in this survey was 24.84 km$^2$. This required a six-person team for one week (42 person days) and 150 h of image analysis time, which at 8 h a day, is 18.75 days. Therefore, the total person days for the drone surveys was 60.75 days. Ground survey teams (consisting of five people) follow line transects (Piel, Cohen et al. 2015), which on standard days, cover 0.176 km$^2$ per day (4 km of transects with a 0.044 km wide estimated strip width (22 m on each side of the transect in woodlands and even less in riparian forests)). Thus, we estimate that ground surveys would require a total of 141 days/person (24.84 ha/0.176 ha) or 705 total person days for a typical team.

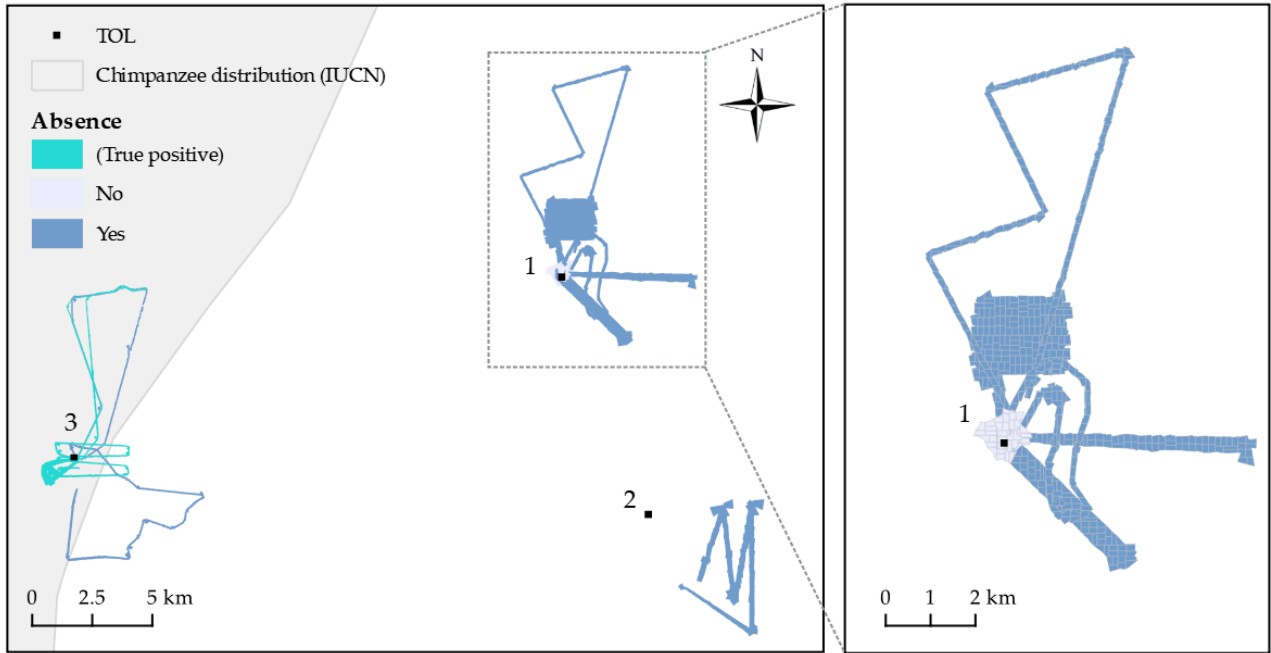

**Figure 4.** This figure shows the flight paths as well as the nest absence estimates. The right panel is a zoom of TOL 1 to show the plots a flight was divided into for the analyses.

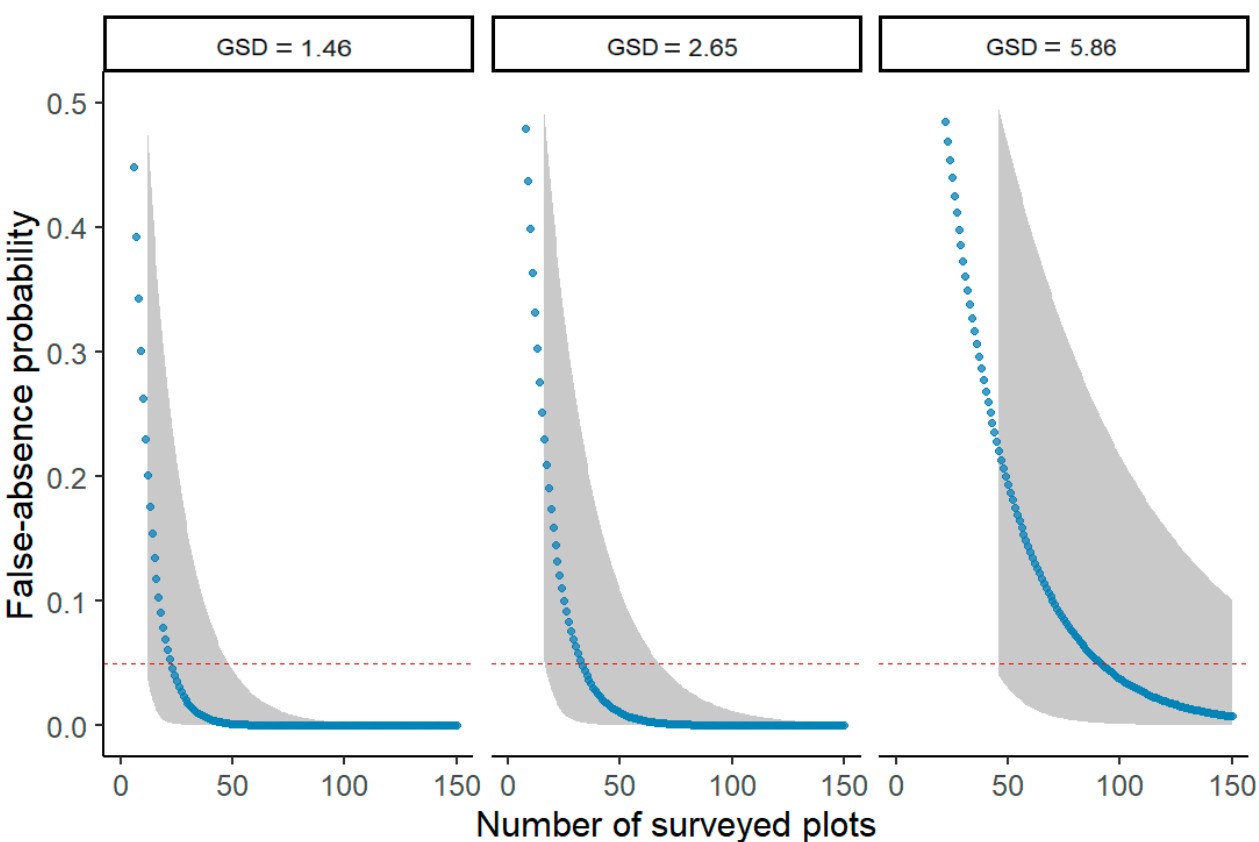

**Figure 5.** Estimated probability of false negative detections during missions where no chimpanzee nests were detected using the formula developed by [37,46]. Estimated probabilities of false negative detections decline with increasing numbers of plots as spatial replicas surveyed during a mission depending on the GSD. Here, we present examples of the lowest GSD (1.46 cm), median GSD (2.65 cm), and highest average GSD (5.86 cm) applied during missions. The red line indicates the 95% probability criterion for a true negative detection.

## 4. Discussion

In this study, we showed that it is feasible to provide a statistical approach to estimate false negative probabilities based on GSD without the use of temporal survey replicas for chimpanzee nest (and by extension, individuals) detection. Substituting temporal for spatial replicas is a valuable approach to account for imperfect detection in drone surveys when logistic or financial restraints do not allow for repeated flights. The resulting absence estimates can aid in improving accuracy of species distribution maps in that species presence rather than species detectability is modelled (Lahoz-Monfort et al., 2014). The presented approach allows for inferring absences based on a selected criterion (95% in our case) by using the formula developed by McArdle (1990) and Kery (2002) and is applicable without the need for temporal replicas.

The area where the surveys were conducted was close to an area in which previous ground survey teams had also detected nests [22], and in three of the flights near where the ground surveys were conducted, we identified 18 nests on the aerial images, which was in line with the expectations [23].

In our study, the false negative probability was influenced by the GSD and the number of plots. The flight that resulted in the highest false negative probability had a GSD of 1.9 cm. For this GSD, 26 plots should have been surveyed to reach a false detection probability of $p = 0.049$. Instead of covering a larger area to obtain more plots, the alternative would have been to fly lower so that detection probability would have increased due to a higher ground sampling distance of the images.

Flying drone missions repeatedly over small areas to infer true absences is often not feasible due to time and money constraints and even less feasible when using other large-scale survey methods such as manned aircraft flights [48]. Instead, we suggest that single survey flights covering large areas can provide spatial replicas to be used for false negative detection estimation as long as replicas are comparable and independent (Kery, 2002). This is the case for surveys of inanimate and hence stationary targets such as nests, as one target cannot appear in multiple spatial replicas. We recommend flying grid missions covering a survey area smaller than the animal home range to ensure that the presence in one plot infers animal (or nest) presence in the complete survey area. A small caveat to our results is that plots that partially consisted of areas that were not suitable chimpanzee habitat (e.g., rivers) and hence were clear areas of true absence were not excluded as spatial replicas. The presence of areas covered by such unsuitable habitat therefore might have slightly inflated real absence estimates in our study. In areas of potential species presence, the only requirement for estimating false negative probabilities is information on the target detection probability per plot of a defined size, which can be inferred from areas of known target abundance. Alternatively, spatial replicas can be used to run hierarchical models that estimate such detection probability as long as numbers of replicas do not differ widely between surveyed areas (MacKenzie et al., 2017). Other attempts to estimate abundance from single flights without prior knowledge of detection probabilities include the use of double-sampling [49] and the use of overlapping images as temporal replicas [50].

The results of our study indicate that drones can be used as a data collection tool to determine the probability of false negatives and thus provide data on presences and absences. Given that species distribution models benefit from both absence and presence data [30], the study indicates the usefulness of drones as a tool for chimpanzee and potentially other great ape studies. There are, however, several challenges. First, nest detection probability at varying GSDs needs to be determined. In our case, we used data from a previous study in an ecologically similar (and nearby) area and assumed that detection probabilities would be similar [23]. If surveys were conducted in different vegetation types where chimpanzees occur—such as dense tropical rainforest—the detection probabilities would likely be different [24]. Second, at present, a human observer is required to interpret the photos, which is an extremely time-consuming process. In our case, the 3560 images required the human image analysts to spend 60 to 30 h on image analyses. To make the process of image analyses faster, machine learning methods that can detect nests automatically would vastly

reduce the analyses time and costs [51,52]. Once such methods are developed, the costs of conducting drone surveys including the subsequent analyses by human image analysts will decline considerably. A critical, additional benefit of aerial imagery is that it produces a form of archival data that can be re-analysed as methods develop in the future, and the data can be repurposed more readily to address additional questions beyond those of the survey designer's intentions. This, itself, adds significant value over traditional approaches. Thus, we believe there is much to be gained from continuing to compare ground and aerial data for a variety of great ape and other nest building species to determine if absences can be determined using drones.

**Supplementary Materials:** The following supporting information can be downloaded at: https://www.mdpi.com/article/10.3390/rs15082019/s1.

**Author Contributions:** Conceptualization, S.A.W. and A.K.P.; data curation, S.A.W., N.B., A.H. and J.T.K.; formal analysis, S.A.W., N.B. and A.H.; funding acquisition, S.A.W. and A.K.P.; investigation, S.A.W., N.B., A.C. and J.T.K.; methodology, N.B., A.H., A.K.P. and J.T.K.; project administration, S.A.W.; resources, L.P.; writing—original draft, S.A.W., N.B. and A.H.; writing—review and editing, S.A.W., N.B., A.H., A.K.P., A.C., F.A.S., L.P. and J.T.K. All authors have read and agreed to the published version of the manuscript.

**Funding:** This study was funded by the National Geographic Society and Liverpool John Moores University. J.K. was also supported by the European Union's Horizon 2020 research and innovation programme under the Marie Skłodowska-Curie grant agreement No 754513 and the Aarhus University Research Foundation.

**Data Availability Statement:** Raw data are available on request to SAW due to dealing with a threatened species. R code for the analyses has been included as a supplementary file.

**Acknowledgments:** We thank the Tanzanian Wildlife Research Institute (TAWIRI), the Tanzanian Commission for Science and Technology (COSTECH), and the Tanganyika District authorities for permission to conduct the work. We are also grateful to the UCSD/Salk Institute Center for Academic Research and Training in Anthropogeny (CARTA) for long-term support to GMERC and the Issa field station.

**Conflicts of Interest:** The authors declare no conflict of interest.

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
