# Peer review of "Using Drones to Determine Chimpanzee Absences at the Edge of Their Distribution in Western Tanzania"

_remotesensing, doi:10.3390/rs15082019_

Round 1

Reviewer 1 Report

This is a short, focused and very well designed methodological paper that implicates the use of drones for chimp nest censusing in a challenging habitat. The authors have done a great job and have analyzed their data with statistical soundness. Moreover, they acknowledge some drawbasks and have highlighted the advantage of the method for current studies, and more interestingly, in future studies. In my opinion the manuscript constitutes an important contribution to those interested in modern censusing techniques in conservation biology. It should be accepted as is. 

Author Response

We thank the reviewer for their kind review of our manuscript and are glad the reviewer liked the manuscript. Given the comments we have nothing to add further in response.

Reviewer 2 Report

Really interesting study about the detection of absent chimpanzee nests using aerial images obtained with a drone. I think the manuscript needs a more careful language and writing, for better clarity. Overall, it is well conducted but in my opinion it lacks a practical application to it, as it is a very time-consuming process and I have doubts if it is really so advantageous to have absences instead of only presences. Moreover, the document does not have line numbers so I could not be more specific in my comments.

Introduction, page 1: As far as I understood, the reference Wilson, Culik et al. 1991 is not about surveying species distribution, but rather understand the impact of man and aircrafts on pinguins.

Page 3: “approximately” repeated twice.

Table 2: is not completely visible. Not necessary to place the “#” – it is better “number”. What are the units of the “probability” columns – is it %? There is no table legend. The “note” part should be in the legend. Acronyms should be discriminated in the legend.

Discussion, page 9: not necessary to disclaim the aim of the study again. Go straight to the interpretation of the results.

So, could we calculate the presences and absences with the same flight images? To decrease the money and time dispended on fieldwork.

I think the paper is missing a comparison between models that have only presences obtained by drones and models that have both the presences and absences. Or any other practical application of having absences of such areas, that require so much effort and time to calculate (60 hours on image processing!). Are there really big advantages of obtaining the absences instead of only the presences? In what cases they can be advantageous? Because there are many studies using only presences or presence-background points with really accurate results.

Author Response

Dear reviewer,

Many thanks for the thoughtful and constructive comments on the manuscript. We have incorporated your suggestions into the new version and provided feedback to the comments in the attached word file.

Best wishes,

Serge

Reviewer 3 Report

Thank you for the opportunity to review "Using drones to determine chimpanzee absences at the edge of their distribution in western Tanzania". The manuscript is reasonably well written overall, however, there are some areas where further definition and refinement is required.  This is the particularly the case when defining flight parameters - which are currently confusing.  There are some standard aviation terminologies that can be used to clear this up and I have given specific examples below.  There are also some minor grammatical errors, also outlined below.  Additionally, some of the figure headings require more information.  These figure headings should be able to be read, and the figure understood, without having to refer back to the manuscript. My major concern with the manuscript is the lack of suitable detail in the methods in particular.  Many aspects are unclear and much further detail is required to both explain and justify why things were done the way they were.   Finally, I would encourage the authors to focus on the title of their manuscript and ensure that the content matches their intent.  The manuscript content could be refined, and the focus of content tightened by this approach.

Introduction

2nd paragraph - should be "...more abundant than..."

3rd paragraph - remove "are" from "...two long-studied populations found in..."

suggest that it should read "...presence and absence of chimpanzee nests..."

Methods

2nd paragraph - why did you fly two types of flight missions?  As far as I can see there is no analysis comparing these two types of missions, so why was it done?  If I was undertaking a search of an area looking for nests, I would think the gridded approach would be the most appropriate.  Why then did you use a linear method?  These linear approaches look haphazard and without structure, and don't look like a formal survey method, (except for the two zigzag approaches in areas 1 and 2).  Further information around survey design is required.

Is it only the gridded missions that had the 60% overlap?  That is what this sentence currently implies. 

Why did you fly at varying altitudes?  Was each flight a different altitude or did you vary altitude within the same flight?  How many replicates of each altitude were there?

repetition "...which approximately took approximately..."

3rd paragraph - during aerial surveys we commonly calculate the amount of ground surveyed by the method as the "swath width x transect length".  My reading is that GSD is the swath width, in which case I would suggest you use this more common terminology.

If it has already been established that flying higher results in lowering the detection probability, then why did you fly at multiple AGL heights?  Your reasoning for this is not clear.

Please don't include results in the Methods section (Table 1, Fig 2 and 3).

4th paragraph - I assume "absolute flight altitude" is referring to Above Mean Sea Level (MSL or AMSL)?  Please use this standard aviation terminology so that others understand what you are referring to.  If it is not AMSL then I can only assume you are referring to Above Ground Level (AGL), which then raises further questions about your methodology regarding calculating your swath width.  Additionally, why did you need to use this method?  Could you not have used AGL from your flight track logs which would give GPS-based AGL data?  If not, why not?  Further detail on why you used this method is required.

I don't understand why you had to use the "lowest flight for each overlapping area".  Could you not just exclude the data from the ferry from the take-off area to the survey block?

Discussion

1st paragraph - You have stated that spatial replicas using drones result in more data more quickly.  This is logical but you didn't compare these two approaches.  This statement needs a reference or should be reframed or removed.

3rd paragraph - now it seems that you are using ground sampling distance in a different way - that the lower your fly the higher the ground sampling distance...which is incorrect based on how you used GSD earlier in the manuscript (as another terminology for swath).  

4th paragraph - you recommend grid missions, yet again I can see no comparison to the linear missions that were flown.  It would seem that the linear missions are not relevant to the manuscript or the results and therefore should be removed.

5th paragraph - there are open-source detection algorithms available that could be adapted to suit your dataset.  I would encourage the authors to look into this further as 180 hours for <4000 images is certainly excessive (as an example I know of a group that uses an algorithm to do a first-pass process of 2500 images in under 30 minutes - they are also looking for round vegetation features).  I also don't agree that using drones and using AI to process imagery are "in their infancy".  There are so many publications on both of these that we are beyond using this terminology in 2023.

Author Response

Dear Reviewer,

Thank you very much for your thoughtful and constructive feedback. We have incorporated your feedback in the new version of the manuscript and provided feedback in the reply word document as well. 

Best wishes,

Serge

Round 2

Reviewer 3 Report

Thank you very much for this revised version.  I have only minor comments

Methods

2nd paragraph - I would actually just remove "approximately" entirely from the sentence.  You use 60 in your time calculations ultimately anyway.

Results

Figure 4 - Please include some information in the figure title regarding linear and gridded flight plans so that the reader understands what they are looking at and why.